# Why It Is Difficult for Military Personnel to Quit Smoking: From the Perspective of Compensatory Health Beliefs

**DOI:** 10.3390/ijerph182212261

**Published:** 2021-11-22

**Authors:** Chor-Sum Au-Yeung, Ren-Fang Chao, Li-Yun Hsu

**Affiliations:** 1Postgraduate Programs in Management, I-Shou University, Kaohsiung 84001, Taiwan; csauyeung0114@gmail.com (C.-S.A.-Y.); energy5544@yahoo.com.tw (L.-Y.H.); 2Department of Leisure Management, I-Shou University, Kaohsiung 84001, Taiwan

**Keywords:** compensatory health beliefs, smoking cessation attitude, smoking cessation motivation, military

## Abstract

Compensatory health beliefs are barriers to healthy behavior. In an effort to understand how the prevalence of these beliefs can be reduced in individuals, 376 valid questionnaires were collected from combat troops in Taiwan. The collected data were analyzed using partial least squares structural equation modelling. It was found that positive attitudes towards smoking cessation had significant negative effects on compensatory health beliefs, while negative attitudes towards smoking cessation significantly enhanced the level of compensatory health beliefs. The motivation for smoking cessation was also found to reinforce the negative effect of positive attitudes towards compensatory health beliefs, while it did not have any significant effect on the relationship between negative attitudes and compensatory health beliefs. Three subconstructs of compensatory health beliefs (exercise, eating habits, and amount of smoking) were found to have simultaneous effects for military personnel. Finally, this study explored the causes of the above-mentioned phenomena, and measures that could reduce the prevalence of compensatory health beliefs were suggested.

## 1. Introduction

Smoking shortens an individual’s life expectancy by an average of 10 years, and each cigarette shortens one’s life expectancy by 15 min [1]. This means that if a soldier smokes, smoking will cause physical problems for the individual and that cumulatively, it will have a profound impact on a country’s military strength. Therefore, the issue of smoking cessation deserves attention. The question of how to enable people to completely quit smoking has been a topic of concern for a long time. Smoking can negatively affect the body and can cause cardiovascular disease and cancer [2]. Lung cancer patients and the general public usually presume the cause of lung cancer to be smoking [3]. Furthermore, smoking has many negative consequences for work, such as increasing one’s vulnerability to injury on the job [4], increasing their risk of mental health problems [5], and increasing the risk of productivity loss, due to poor health [6]. Therefore, quitting smoking is of great importance, for the positive effects it has on both smokers’ health and their quality of work.

Smokers are typically aware of the disadvantages of smoking and understand the health and work benefits of quitting. So why are they unable to quit smoking altogether? Many medical studies have suggested that the nicotine contained in cigarettes can quickly cause addiction that is difficult to overcome [7], making it easy for those who try to quit to resume smoking. The present study was undertaken to explore the difficulties of quitting smoking from a psychological perspective.

Past research regarding the question of how to encourage healthy habits, such as increasing the willingness to eat fruits and vegetables [8], or how to quit unhealthy habits, such as the consumption of unhealthy snacks [9,10], has commonly approached the interpretation of their findings from the perspective of compensatory health beliefs (CHBs). CHBs refer to the idea that people can compensate for or even cancel out poor behaviors through positive behaviors, such as frequently exercising, eating healthily, or smoking less [11]. For example, smokers who drink less alcohol think that this behavior can compensate for the disadvantages of smoking [12]. Studies have shown that CHBs can be used as an excuse to indulge in unhealthy behaviors; Fuentes & Almagiá [13] found that there is a negative correlation between CHBs and health, so they can lead to increased alcohol consumption [14] and smoking [15]. These findings show that smokers with more CHBs are less likely to quit smoking because they use these beliefs to justify their smoking behavior, which leads to a reduced willingness to quit [16]. CHBs can explain why smokers believe that smoking is acceptable [11] and can help to identify a person’s motivation to quit smoking (MO) [13], which is an obstacle to successful quitting. This study focused on the issue of how to prevent smokers from forming CHBs and help them quit smoking successfully.

Based on the importance of smoking cessation for military personnel and the lack of understanding surrounding smoking cessation, the purpose of this study was to explore the degree of the influence of the attitude towards smoking cessation (positive and negative) on CHBs and, in the context of MO, the impact of the change in the degree of smoking cessation attitude (positive and negative) on CHB formation.

## 2. Literature Review

### 2.1. Why Is It More Challenging for Taiwanese Soldiers to Quit Smoking?

Previous studies have proved that soldiers work in stressful environments, so they are prone to post-traumatic stress disorder (PTSD), which increases the chances and risks of smoking for soldiers [17,18]. In addition, military personnel describe the use of tobacco as a way to reduce and manage stress, anger, and boredom [19], meaning military personnel are more likely to smoke than civilians [20]. Smoking is a common health problem faced by military personnel all over the world (e.g., in Uganda, USA, and Australia). Military personnel in Taiwan also commonly experience problems regarding quitting smoking.

Due to the unique nature of the work of Taiwan’s military personnel, daily life in the military is standardized, and individuals are easily influenced by peers [12]. Military personnel often face safety threats that lead to stress, anxiety, and reduced sleep. In addition, they are often bored during leisure time [21] and may smoke to relax or cope with negative emotions [22]. Smoking is thus a part of military culture [7]; Reitsma et al. [23] pointed out that the smoking rates in militaries around the world are as high as 20–66%, implying that nearly half of military personnel worldwide have a smoking habit. In Taiwan, the path to becoming a service member usually begins by studying at a military school, enlisting in the national army, or by fulfilling obligatory military service as an adult. These pathways have led to a trend of a larger proportion of younger people within Taiwan’s military. Awareness of the consequences of smoking and attitudes towards the habit are significantly related to whether an individual smokes, so younger military personnel may have a lower level of understanding and more passive attitudes towards smoking [2]. With an already existent smoking culture in the military, young recruits have a higher smoking rate than the general public [24]. Although Taiwan’s military has introduced various smoking cessation programs, and the smoking rate among military personnel has decreased slightly [25], it has remained above 30% [12]. Furthermore, London et al. [7] pointed out that most military personnel continue to smoke after leaving the military, which may cause veterans to develop long-term tobacco addictions that affect their health. Questions that remain unanswered include the question of why smoking is so prevalent in Taiwan’s military, as well as the issue of why it is so difficult for military personnel to quit smoking.

These questions can be tackled by considering CHBs. The concept of CHBs has been applied in various fields for a long time, especially in the field of health medicine. Most jobs in the military are physically demanding, and military personnel exercise more than an average person. As a result, they may develop the misconception that the amount of physical activity and exercise they do can compensate for the lung damage caused by smoking. In the military, diet is also strictly controlled and is considered relatively healthy and nutritionally balanced. Rules and regulations in the military are quite rigorous, making it difficult to consume alcohol, thereby eliminating another factor that negatively affects health. These considerations are in line with the three subconstructs of CHBs that have been proposed by Radtke et al.: exercise (CE), eating habits (EH), and amount of smoking (AS) [15]. Therefore, it is reasonable to evaluate the CHBs of Taiwan’s military personnel with these three constructs.

CHBs cannot actually offset the negative consequences caused by unhealthy behaviors, and compensation behaviors are not effectively implemented [13]. In other words, CHBs are therefore “self-deception” concepts. Therefore, this study aimed to find the factors that constitute CHBs from a psychological level, so as to reduce CHBs, and even prevent their occurrence.

### 2.2. What Factors Can Affect the Formation of CHBs?

Previous studies have suggested that an individual’s level of self-concordance, their degree of desirability, their self-efficacy, and their motivational conflict responses may be among the factors that constitute CHBs [26]. In addition, Thongworn and Sirisuk [26] found that some smokers were less concerned about the adverse consequences of smoking. In other words, they had a negative attitude (NA) towards smoking cessation, which can also contribute to CHBs. Conversely, when military personnel have positive attitudes (PAs) towards smoking cessation, they may be less likely to use the physical demands and controlled diet of their job as excuses for smoking. As positive and negative attitudes are two extremes, it is necessary to discuss them separately; to answer the proposed questions, the roles of these attitudes in smoking cessation were a major focus of this study.

The MO towards smoking cessation is also a critical factor for successfully quitting. When military personnel are motivated to quit, this could affect the relationship between their attitudes and CHBs. Indeed, MO has been determined to be a critical factor that affects attitudes [27,28]. The higher their MO for smoking cessation, the more that smokers want to achieve their desired goal; thus, they would be more determined to quit, and less likely to take actions to compensate for smoking. Indeed, Radtke and Rackow [29] found that the MO for smoking cessation affected CHBs. Consequently, this MO can be viewed as a situational factor that can be used to further explore how the relationship between attitudes and CHBs may change.

## 3. Methods

### 3.1. Study Design

Figure 1 presents the research framework, which surrounds the following four hypotheses: (1) PAs towards smoking cessation have a negative effect on CHBs; (2) NAs towards smoking cessation have a positive effect on CHBs; (3) The MO for smoking cessation is considered a PA, which has a negative moderating effect on CHBs; (4) The MO for smoking cessation is considered an NA, which has a positive moderating effect on CHBs. Hopefully, the results from testing these hypotheses can help to answer the question of why it is so difficult for Taiwanese military personnel to quit smoking, as well as provide practical suggestions for Taiwan’s military to help prevent personnel from developing CHBs.

The hypotheses of this study were as follows:1.PAs towards smoking cessation have a negative effect on CHBs.

**Hypothesis** **1.***PAs towards smoking cessation have a negative effect on CE*.

**Hypothesis** **2.***PAs towards smoking cessation have a negative effect on EH*.

**Hypothesis** **3.***PAs towards smoking cessation have a negative effect on AS*.

2.NAs towards smoking cessation have a positive effect on CHBs.

**Hypothesis** **4.***NAs towards smoking cessation have a positive effect on CE*.

**Hypothesis** **5.***NAs towards smoking cessation have a positive effect on EH*.

**Hypothesis** **6.***NAs towards smoking cessation have a positive effect on AS*.

3.The intensity of the MO for smoking cessation is a PA, which has a negative moderating effect on CHBs.

**Hypothesis** **7.***The intensity of the MO for smoking cessation has a negative moderating effect on CE*.

**Hypothesis** **8.***The intensity of the MO for smoking cessation has a negative moderating effect on EH*.

**Hypothesis** **9.***The intensity of the MO for smoking cessation has a negative moderating effect on AS*.

4.The intensity of the MO for smoking cessation is an NA, which has a positive moderating effect on CHBs.

**Hypothesis** **10.***The intensity of the MO for smoking cessation has a positive moderating effect on CE*.

**Hypothesis** **11.***The intensity of the MO for smoking cessation has a positive moderating effect on EH*.

**Hypothesis** **12.***The intensity of the MO for smoking cessation has a positive moderating effect on AS*.

### 3.2. Participants

Combat troops from Taiwan’s military were selected as the study population because their lifestyle best reflects the characteristics of military personnel. We collected a total of 447 participants who were successfully recruited from 2 May 2021 to 8 June 2021. After deducting incomplete questionnaires, data from a total of 376 participants (346 men and 30 women) were included in the data analysis. Most of the participants interviewed were 21–25 years old (*n* = 233; 61.97%). In total, 167 participants (44.41%) had university degrees, and 128 (34.04%) had an education level of high school education or lower. Almost half the participants had a smoking history of five years or less (*n* = 170; 45.21%), and 43.88% (*n* = 165) had smoked for 6–10 years. A total of 181 participants (48.14%) had served in the military for less than six months. In terms of rank, 73.14% (*n* = 275) were soldiers. Most participants (*n* = 247; 65.60%) had thought about quitting smoking. The sample profile is provided in Table 1.

### 3.3. Questionnaire Design

The questionnaire used in this study was divided into four parts. In the first part, the scale that was proposed by Etter et al. [30] was used to evaluate the attitudes towards smoking cessation. This scale was originally divided into three subconstructs, including the adverse effects of smoking, the psychoactive benefits of smoking, and the pleasure of smoking. Because the focus of this study was on positive and negative attitudes, the awareness about the adverse effects of smoking was classified as a PA, and the psychoactive benefits and pleasure of smoking were combined and considered to be NAs. The second part of the questionnaire adopted the scale that was proposed by Radtke et al. [15] to evaluate the CHBs of military personnel, where three subconstructs were investigated: CE, EH, and AS. The third part of the questionnaire adopted the scale that was proposed by Cupertino et al. [31] to evaluate the MO for smoking cessation. The narratives accompanying the scales that were used were adjusted in accordance with the needs of this study. The last part of the questionnaire included basic information about the participants. The items of each construct are displayed in Appendix A.

### 3.4. Statistical Method

In this study, partial least squares structural equation modelling (PLS-SEM) was adopted for use in the data analysis using SmartPLS 3.0 (SmartPLS GmbH, Boenningstedt, Germany). This method was used to simultaneously evaluate the relationships between research framework constructs [32], and could also be used to assess moderating effects more accurately [33]. To ensure the precision of the data analysis, it was divided into two parts, based on the recommendation by Chin [34]. The first part involved analyzing and evaluating whether the collected data were credible: a confirmatory factor analysis (CFA) was used to determine the reliability and validity of each questionnaire item. In the second part, a path analysis was carried out, to determine whether there was a significant effect.

## 4. Results

### 4.1. Measurement Model

To determine whether the collected data were normally distributed, skewness and kurtosis values of ±2 were used [35]. The skewness of all items was found to be between −1.095 and 0.032, while the kurtosis was between −1.097 and 1.018 for all items. These values fell within the given range, indicating that the data collected met the criteria of normal distribution.

To verify the validity and reliability of the collected data, the factor loading, the composite reliability (CR), and the average variance extracted (AVE) values were used as the criteria. According to Hair Jr. et al. [36], factor loading values above 0.6, CR values above 0.7, and AVE values above 0.5 indicate a high convergence validity. Table 2 presents the CFA results and shows that the values of all items met the recommended thresholds [36], which indicated that the variables in this study met the standards required of the measurement model.

According to Kline [37], an absolute value of the correlation coefficient between constructs less than 0.850 implies discriminant validity. As shown in Table 3, the absolute values ranged from 0.026 to 0.731, confirming discriminant validity between the constructs. In addition, Hair Jr. et al. [36] pointed out that an absolute value of the correlation coefficient less than the square root of the AVE value indicates good discriminant validity. The absolute values for all constructs met this requirement, reaffirming that the data that were collected in this study had good discriminant validity.

### 4.2. Structural Model

A path analysis was performed by calculating the coefficient of each hypothetical path, using the maximum likelihood method to verify the relationships and influences of the study model. Table 4 summarizes the hypothesis verification results of the theoretical framework. All the paths were statistically significant except for H10–12, because smoking cessation MO had no significant positive moderating effect on negative attitudes towards smoking cessation and CHBs.

## 5. Discussion

### 5.1. Predicting the Influence of Attitudes towards Smoking Cessation on CHB

Although many previous studies have evaluated smokers’ attitudes towards smoking cessation [38,39] and CHBs [9,16], predictions regarding the influence of these two behaviors on each other have been neglected. Studies have evaluated issues related to smoking cessation from the perspective of attitude [39,40], but there has also been a lack of research that has focused on the two types of attitudes (positive and negative) towards smoking cessation. The results of the current research help to fill these gaps.

Attitudes towards smoking cessation often reveal the reasons why people resume smoking [30,38]. When individuals exhibit NAs, they are more likely to develop the idea of resuming smoking, which, in turn, creates a psychological conflict between smoking and quitting. Under this circumstance, CHBs form that give smokers excuses to continue smoking and reduce their psychological conflicts [16]. Increasing individuals’ awareness of the negative effects of smoking can reduce the generation of CHBs [29]. When military personnel have more PAs towards smoking cessation and recognize that smoking is wrong, they are more likely to avoid forming CHBs and are less likely to experience psychological conflicts between smoking and quitting [16].

Radtke et al. [15] determined that the CE, EH, and AS were three important subconstructs of CHBs. Theoretically, the three subconstructs do not need to coincide, because any one of them can alleviate psychological conflict. This is also the reason why the three subconstructs were analyzed separately in this study. Based on the results, a PA towards smoking cessation had simultaneous negative effects on the CE, EH, and AS, and vice versa. Military personnel are often susceptible to peer influence due to the unique nature of their work and standardized lifestyle [13]. The physical demands of their job, their diet, and even changes in the amount they smoke are all susceptible to peer influence, which leads to a phenomenon of interconnectivity among the three subconstructs. In other words, the CE, EH, and AS of military personnel that smoke were simultaneously affected.

### 5.2. The Moderating Role of Smoking Cessation MO

MO is a process that initiates, controls, strengthens, or maintains one’s behavior [41] and is the driving force behind the actions of humans. With a high level of MO for smoking cessation, there is a stronger determination to quit, and the PAs towards smoking cessation are maintained. Individuals are less likely to have psychological conflicts between smoking and quitting and are less likely to use a CHB as an excuse to continue smoking.

Miquelon et al. [42] pointed out that people with a higher MO had fewer CHBs; a higher MO may be associated with a stronger resistance to temptation and a more robust commitment to goals [43]. When smokers are motivated to cease smoking, they exhibit goal-achieving behavior and are more capable of resisting the desire to smoke, which interrupts the formation of CHBs [42]. This present study demonstrated that, with a higher MO for smoking cessation, individuals have fewer CHBs, provided that they have a PA towards quitting. When people held NAs towards smoking cessation, their CHBs were not affected by their MO. When military personnel were highly motivated to quit smoking, they were more willing to cope with the urge to smoke [41] and would not use a CHB as an excuse for smoking. On the other hand, most people with NAs towards smoking cessation had no intention of quitting, or only adopted passive methods to quit. Even in an environment that was conducive to quitting, they would still not change their mind or behavior.

### 5.3. Research Implications

From past research, it was found that soldiers encounter difficulties in quitting smoking, and even after leaving the army, they still use smoking to cope with negative emotions [44]. This study found that the above reasons are caused by CHBs, and that even the smoking cessation attitude and smoking cessation MO of an individual can affect the occurrence of CHBs.

According to the results of this study, it was found that the reason why it is difficult for military personnel to quit smoking is because, cognitively, they feel that they can compensate through other methods. The previous literature showed that this compensatory psychology will not slow down the harm caused by smoking [11]. This is the reason why it is currently difficult for soldiers to quit smoking. Therefore, health units in the military must help soldiers form the correct cognition in this regard, through methods such as establishing a supportive environment for smoke prevention and control and the strengthening of officers’ and soldiers’ awareness of the dangers of smoking.

## 6. Conclusions

The results of this study show that PAs towards smoking cessation have a significant negative effect on the three subconstructs of CHBs (CE, EH, and AS), while NAs have a significant positive effect on these subconstructs. These results indicate that those military personnel with more PAs towards smoking cessation are less likely to try to make up for the harms of smoking through CHBs. In contrast, those military personnel with more NAs towards smoking cessation believed that corrective actions could compensate for the disadvantages of smoking. Furthermore, this study demonstrated that the three subconstructs of CHBs were simultaneously affected, and therefore, moderating effects exist among them. Lastly, the MO for smoking cessation, which is considered to be a PA, had negative moderating effects on the three subconstructs, while NAs showed no moderating effects. This finding implies that although the MO significantly affects CHBs, its impact was limited to those military personnel with PAs, rather than NAs, towards smoking cessation.

The final conclusion of this study is that as both attitude and MO can affect the CHBs of soldiers; we believe that CHBs are psychological problems. In fact, CHBs not only affect current unhealthy behaviors but also affect treatment compliance [15]; in other words, treatment compliance may be affected by CHB factors, which may affect the best time for treatment, or even cause the best time for treatment to be missed. Therefore, CHBs have a certain degree of negative influence on medical treatment or medicine.

### 6.1. Practical Recommendations

In terms of practical recommendations based on these results, programs should be developed to help enhance the PA of military personnel towards smoking cessation, while discouraging NAs. For example, establishing a supportive environment for smoking hazard prevention and control, as well as encouraging awareness of the harms that are associated with smoking, are two ways to cultivate PAs towards smoking cessation and reduce CHBs. In addition, the MO for smoking cessation was only found to affect those with PAs, so more counselling opportunities, such as lectures and psychological counselling, should be provided to personnel who are willing to quit smoking.

### 6.2. Limitations

This research only used combat troops in Taiwan’s military as the research object, although this causes research restrictions. However, due to the characteristics of the military, the military situation in each country or region is actually similar to each other. Even though the target of this study was only the combat force in Taiwan, it is still credible for use in discussing the impact of CHBs in the military in other countries. In particular, this study was based on combat troops, which can better demonstrate the effect of CHBs on smoking cessation with regard to the combat characteristics of soldiers.

### 6.3. Future Research

The three subconstructs of CHBs were found to coincide, which was possibly due to the unique nature of military work and its standardized lifestyle [11]. Future research should examine other occupational groups to determine whether there is any connectivity among CE, EH, and AS, to further verify the constituent elements of CHBs.

With regard to CHBs in relation to smoking, the existence of designated smoking areas may also be one factor that promotes smoking in the military [45]. In other words, there are actually many external factors that cause soldiers to smoke. Ultimately, this is also worthy of further research.

## Figures and Tables

**Figure 1 ijerph-18-12261-f001:**
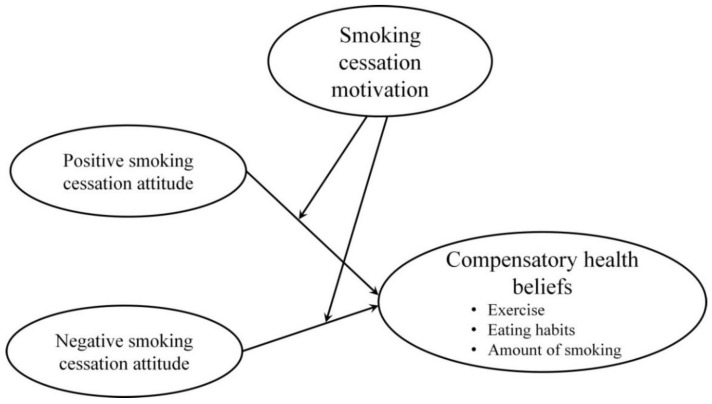
Research model.

**Table 1 ijerph-18-12261-t001:** Profile of respondents (*n* = 376).

Categories	Variables	*n*	%
Gender	Male	346	92.02
	Female	30	7.98
Age	Under 20	43	11.44
	21–25	233	61.97
	26–30	64	17.02
	31–35	30	7.98
	36–40	3	0.80
	46 and over	3	0.80
Education level	High school education or lower	128	34.04
	College degrees	49	13.03
	University degrees	167	44.41
	Postgraduate and over	32	8.51
Smoking history	Under 5 years	170	45.21
	6–10 years	165	43.88
	11–20 years	37	9.84
	21 years and over	4	1.06
Military time	Under six months	181	48.14
	1 year	31	8.24
	1–2 year(s)	54	14.36
	3–5 years	44	11.70
	6–10 years	44	11.70
	11–15 years	20	5.32
	16 years or over	2	0.53
Rank	Soldiers	275	73.14
	Sergeant	80	21.28
	Officer	21	5.59
Had thought about quitting smoking	Yes	247	65.69
	No	129	34.31

**Table 2 ijerph-18-12261-t002:** Confirmatory factor analysis and scale reliability.

Constructs	Items	Loading	CR	AVE
Attitude				
Positive	PA1	0.783	0.959	0.703
	PA2	0.842		
	PA3	0.869		
	PA4	0.824		
	PA5	0.809		
	PA6	0.878		
	PA7	0.845		
	PA8	0.866		
	PA9	0.845		
	PA10	0.818		
Negative	NA1	0.786	0.957	0.738
	NA2	0.873		
	NA3	0.863		
	NA4	0.887		
	NA5	0.821		
	NA6	0.911		
	NA7	0.872		
	NA8	0.853		
CHB				
CE	CE1	0.946	0.954	0.874
	CE2	0.938		
	CE3	0.919		
EH	EH1	0.834	0.917	0.733
	EH2	0.889		
	EH3	0.864		
	EH4	0.837		
AS	AS1	0.883	0.924	0.802
	AS2	0.896		
	AS3	0.907		
Smoking cessation MO	MO1	0.826	0.960	0.751
	MO2	0.840		
	MO3	0.857		
	MO4	0.854		
	MO5	0.904		
	MO6	0.880		
	MO7	0.880		
	MO8	0.889		

**Table 3 ijerph-18-12261-t003:** Discriminant validity assessment.

Constructs	PA	NA	CE	EH	AS	MO
PA	0.838					
NA	−0.031	0.859				
CE	−0.097	0.213	0.935			
EH	−0.080	0.443	0.355	0.856		
AS	−0.160	0.376	0.300	0.390	0.895	
MO	0.731	−0.167	−0.026	−0.053	−0.152	0.867

Note: Abbreviations: PA is positive attitude, NA is negative attitude, CE is exercise, EH is eating habits, AS is amount of smoking, and MO is smoking cessation motivation. Diagonal elements (shaded) are the square roots of the average variance extracted (AVE) values.

**Table 4 ijerph-18-12261-t004:** Significance of hypotheses and validation results.

Hypothesis	Coefficient	t	*p*	Supported
1. PA → CHB				
H1: PA → CE	−0.193	2.140	0.032	Y
H2: PA → EH	−0.176	2.169	0.030	Y
H3: PA → AS	−0.144	1.992	0.046	Y
2. NA → CHB				
H4: NA → CE	0.188	3.110	0.002	Y
H5: NA → EH	0.449	7.666	<0.001	Y
H6: NA → AS	0.305	5.910	<0.001	Y
3. Moderating effect of MO between PAs and CHBs				
H7: MO × PA → CE	−0.223	3.929	<0.001	Y
H8: MO × PA → EH	−0.184	3.455	0.001	Y
H9: MO × PA → AS	−0.232	4.769	<0.001	Y
4. Moderating effect of MO between NAs and CHBs				
H10: MO × NA → CE	0.017	0.331	0.740	N
H11: MO × NA → EH	0.020	0.407	0.684	N
H12: MO × NA → AS	0.052	1.186	0.236	N

Note: Abbreviations: PA is positive attitude, NA is negative attitude, CE is exercise, EH is eating habits, AS is amount of smoking, and MO is smoking cessation motivation. The standard of significance is based on *p* < 0.05.

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
