# Peer review of "Why It Is Difficult for Military Personnel to Quit Smoking: From the Perspective of Compensatory Health Beliefs"

_ijerph, 2021, doi:10.3390/ijerph182212261_

Round 1

Reviewer 1 Report

Title: The question in the title is not answered in the article.

Introduction: a broader international context is missing; line 30: delete preposition "by" before "increasing".

Literature Review: Research framework in Figure 1 and study hypotheses in line from 123 to 154 should be a part of study methods. It would be advisable to include this part of study as separate subchapter „Study design“. Also, It would be appropriate to draw study hypotheses into a block diagram for better clarity for the reader.

line 80:  reword part "the lung damage smoking causes".

line 113-116: 3rd and 4th hyptheses are not fully clear, it seems that motivation for smoking cessation is considered a positive and negative attitude at the same time, it is confusing, try to rewrite.

Methods (3.1. Questionnaire Design): A sample of questionnaire should be available in the article, e.g. as an appendix to the article. It would be useful to indicate the exact questions that the study participants were confronted with.

Methods (3.2. Participants): Please explain the duration of study period mentioned in line 173.

Methods (3.3. Statistical method): There is no mention of the level of significance used in the statistical analysis. Statistical significance is usually set at p < 0.05.

Results: Splitting Table 2 into two parts is unnecessary. I suggest combining the data into one complete table of confirmatory factor analysis and scale reliability.

Results:  You should explain each of your abbreviations the first time it appears in the main text. Specifically the abbreviations (PA; NA; CE; EH; AS; MO) should be once explained in Table 2 (or in associated text near this table). Subsequently, the abbreviation should always be used in the rest of the manuscript instead of the complete term (e.g. in Table 3).

Conclusion: From this chapter it is not clear why it is difficult to quit smoking, authors mainly focus on positive attitudes and how to enhance them. Does the research claim, that due to specific conditions in military services, personel do not see reason for quitting smoking, or is not willing to quit, or is too influenced by "peers"? What the authors arrived at and what is it important for the further direction of the given "medical problem"

Author Response

Dear Editor and Reviewers,

Once again, thank you very much for the comments and suggestions on our revised manuscript. We are grateful to be given the opportunity to submit our revised manuscript, and to respond to some final-stage concerns. We have carefully considered the reviewers’ suggestions and advice, and made further improvements to the manuscript.  Responses to each individual comment are contained in the table below.

We have responded to the reviewers accordingly below and highlighted the changes in yellow background in the manuscript.

In addition, in order to make the English usage of the manuscript meet the requirements of academic journals, the manuscript has been edited by the English editing service provided by MDPI (English-Editing-Certificate-36901). The use of English in the manuscript should meet academic requirements.

Kind regards,
The Authors

Reviewer 1
[Comments] Title: The question in the title is not answered in the article.
[Response] Thanks. This is a good suggestion. We have added a subsection of "5.3 Research implications". This subsection not only summarizes the significance of this research, but also answers the reasons why it is difficult for soldiers. The results show that it is caused by compensatory health beliefs and will be affected by attitudes and motivations. Please see line 292-304.

[Comments] Introduction: a broader international context is missing; line 30: delete preposition "by" before "increasing".
[Response] Thanks for the reviewer's suggestion. We have increased the commonality of military smoking on international health issues, and emphasized the role of compensatory health beliefs. Please see line 58-60, and line 70-77. In addition, we have deleted redundant prepositions as suggested by the reviewer.

[Comments] Literature Review: Research framework in Figure 1 and study hypotheses in line from 123 to 154 should be a part of study methods. It would be advisable to include this part of study as separate subchapter „Study design“. Also, It would be appropriate to draw study hypotheses into a block diagram for better clarity for the reader.
[Response] Thanks for the reviewer's suggestion. We have moved the research structure and research hypothesis to the new subsection of "3.1 Study design". Please see line 138-169.

[Comments] Line 80:  reword part "the lung damage smoking causes".
[Response] Thanks for the reviewer's suggestion. We have corrected the description. Please see line 103-104.

[Comments] Line 113-116: 3rd and 4th hypotheses are not fully clear, it seems that motivation for smoking cessation is considered a positive and negative attitude at the same time; it is confusing, try to rewrite.
[Response] Thanks. The motivation to quit smoking as a moderation role is an external situational factor of intensity. We tried to understand the influence of military personnel's motivation to quit smoking on the relationship between attitudes and CHB. To avoid misunderstandings, we revised the description of the hypothesis. Please see line 157-166.

[Comments] Methods (3.1. Questionnaire Design): A sample of questionnaire should be available in the article, e.g. as an appendix to the article. It would be useful to indicate the exact questions that the study participants were confronted with.
[Response] Thanks for the reviewer's suggestion. We provide the questionnaire items and reference sources used in this study in the appendix at the end of the manuscript.

[Comments] Methods (3.2. Participants): Please explain the duration of study period mentioned in line 173.
[Response] Thanks. This may be a misunderstanding in the description. We have made the following amendments to this sentence: “We collected a total of 447 participants who were successfully recruited from 2021/5/2 to 2021/6/8.” Please see line 172-173.

[Comments] Methods (3.3. Statistical method): There is no mention of the level of significance used in the statistical analysis. Statistical significance is usually set at p < 0.05.
[Response] Thanks for the reviewer's suggestion. The significance standard we use is indeed p<.05. The description of this standard is added below Table 4. Please see line 245-246.

[Comments] Results: Splitting Table 2 into two parts is unnecessary. I suggest combining the data into one complete table of confirmatory factor analysis and scale reliability.
[Response] Thanks for the reviewer's suggestion. We have adjusted Table 2 appropriately.

[Comments] Results:  You should explain each of your abbreviations the first time it appears in the main text. Specifically the abbreviations (PA; NA; CE; EH; AS; MO) should be once explained in Table 2 (or in associated text near this table). Subsequently, the abbreviation should always be used in the rest of the manuscript instead of the complete term (e.g. in Table 3).
[Response] Thanks for the reviewer's suggestion. In order to make the article more concise, we have presented the key terms of the whole article in abbreviations and explained them when they first appeared.

[Comments] Conclusion: From this chapter it is not clear why it is difficult to quit smoking, authors mainly focus on positive attitudes and how to enhance them. Does the research claim, that due to specific conditions in military services, personnel do not see reason for quitting smoking, or is not willing to quit, or is too influenced by "peers"? What the authors arrived at and what is it important for the further direction of the given "medical problem"?
[Response] Thanks for the reviewer's suggestion. This study mainly explores the reasons why soldiers encounter difficulties in quitting smoking. The main focus of this research is on the psychological level. In other words, CHB will be the reason why it is difficult for soldiers to quit smoking. In addition, attitudes and motivations will affect the CHB of soldiers. In order to strengthen this concept, this article adds a further description. Please see line 318-323.

Reviewer 2 Report

It is an paper that has important theoretical and applied contributions in relation to smoking behavior and its cessation, as well as complying with the general guidelines established by the journal en terms of format. However, it is necessary to:

-Clearly state the purpose of the study.

-Review the order of the content of the Method: first are the participants and then the Questionnaire design section.

-It is suggested that the hypotheses be included as specific purpose.

- In the discussion it is suggested to include the limitations of the study and the proposals for future research.

Author Response

Dear Editor and Reviewers,

Once again, thank you very much for the comments and suggestions on our revised manuscript. We are grateful to be given the opportunity to submit our revised manuscript, and to respond to some final-stage concerns. We have carefully considered the reviewers’ suggestions and advice, and made further improvements to the manuscript.  Responses to each individual comment are contained in the table below.

We have responded to the reviewers accordingly below and highlighted the changes in yellow background in the manuscript.

In addition, in order to make the English usage of the manuscript meet the requirements of academic journals, the manuscript has been edited by the English editing service provided by MDPI (English-Editing-Certificate-36901). The use of English in the manuscript should meet academic requirements.

Kind regards,
The Authors

Reviewer 2
[Comments] Clearly state the purpose of the study.
[Response] Thanks for the reviewer's suggestion. We clearly stated the purpose of this research in the "Introduction" section. Please see line 63-67.

[Comments] Review the order of the content of the Method: first are the participants and then the Questionnaire design section.
[Response] Thanks for the reviewer's suggestion. We have adjusted the order of subsections according to the reviewer's suggestion.

[Comments] It is suggested that the hypotheses be included as specific purpose.
[Response] Thanks for the reviewer's suggestion. We have linked the hypothesis and purpose appropriately. Please see line 63-67.

[Comments] In the discussion it is suggested to include the limitations of the study and the proposals for future research.
[Response] Thanks for the reviewer's suggestion. We have added two subsections "6.2 Limitation" and "6.3 Future researches". Please see line 333-349.

Round 2

Reviewer 1 Report

All my comments have been processed successfully.